# *In vitro* immune-enhancing effects of *Platycodon grandiflorum* combined with *Salvia plebeian* via MAPK and NF-κB signaling in RAW264.7 cells

A-yeong Jang[1,2☯], Minji Kim[1,3☯], Weerawan Rod-in[1,4☯], Yu Suk Nam[5], Tae Young Yoo[6], Woo Jung Park[1,2]*

1 Department of Marine Bio Food Science, Gangneung-Wonju National University, Gangneung, Gangwon, Korea, 2 Department of Food Science and Technology, Gangneung-Wonju National University, Gangneung, Gangwon, Korea, 3 Department of Wellness-Bio Industry, Gangneung-Wonju National University, Gangneung, Gangwon, Korea, 4 Department of Agricultural Science, Faculty of Agriculture Natural Resources and Environment, Naresuan University, Phitsanulok, Thailand, 5 NAAAMYUUU FNC Co., Ltd, Seoul, Korea, 6 FD FARM Co., Ltd, Icheon, Gyeonggi-do, Korea

☯ These authors contributed equally to this work.
* pwj0505@gwnu.ac.kr

**Data Availability Statement:** All relevant data are within the manuscript and its Supporting Information files.

## Abstract

The immune-enhancing activity of the combination of *Platycodon grandiflorum* and *Salvia plebeian* extracts (PGSP) was evaluated through macrophage activation using RAW264.7 cells. PGSP (250–1000 μg/mL) showed a higher release of NO in a dose-dependent manner. The results showed that PGSP could significantly stimulate the production of $PGE_2$, COX-2, TNF-α, IL-1β, and IL-6 in RAW264.7 cells and promote *iNOS*, *COX-2*, *TNF-α*, *IL-1β*, *IL-4*, and *IL-6* mRNA expression. Western blot analysis demonstrated that the protein expression of iNOS and COX-2 and the phosphorylation of ERK, JNK, p38, and NF-κB p65 were greatly increased in PGSP-treated cells. PGSP also promoted the phagocytic activity of RAW264.7 cells. All these results indicated that PGSP might activate macrophages through MAPK and NF-κB signaling pathways. Taken together, PGSP may be considered to have immune-enhancing activity and might be used as a potential immune-enhancing agent.

## Introduction

Immunostimulatory agents are used to enhance non-specific immune responses and regulate immunity to protect the body [1]. Numerous natural compounds have been shown to possess immunostimulant properties [1–3]. Several immunomodulators released by macrophages include nitric oxide (NO), prostaglandin E2 ($PGE_2$), inducible nitric oxide synthase (iNOS), cyclooxygenase-2 (COX-2), interleukin-1β (IL-1β), IL-4, IL-6, IL-10, IL-12, monocyte chemoattractant protein-1 (MCP-1), and tumor necrosis factor-α (TNF-α), which enhance immune responses [1, 3–6]. Furthermore, the activation of macrophages for the production of

**Funding:** This study was supported by a research program grant funded by NAAAMYUUU FNC Co., Ltd. and FD Farm Co., Ltd. The funders had no role in study design, data collection and analysis, decision to publish, or preparation of the manuscript.

**Competing interests:** The authors have declared that no competing interests exist.

immunomodulators was also reported to be mainly promoted by nuclear factor-κB (NF-κB) and mitogen-activated protein kinase (MAPK) signals [1–3, 5].

The balloon flower, *Platycodon grandiflorum* (Jacq.) A. DC., belonging to the Campanulaceae family, is found in China, Korea, Japan, Russia, and Siberia [7]. It has been recognized as a functional source and traditional oriental medicine [8, 9]. According to reports on phytochemical studies, *P. grandiflorum* contains triterpenoid saponins (platycodin D, platycoside E, platyconic acid A, and deapioplatycoside E), phenolic acids, minerals, volatile oils, polyacetylene, flavonoids, and polysaccharides that are health-promoting [7]. Pharmacological studies showed that *P. grandiflorum* had anti-hypercholesterolemic [10], anti-cancer [9, 11], anti-proliferative [12], anti-obesity [10], anti-atherosclerotic [13], antioxidant [8, 9, 14], immunomodulatory [15], anti-inflammatory [14, 16], immuno-enhancing [17, 18], and anti-microbial [19] properties in *in vitro* and *in vivo* models. Polysaccharides from *P. grandiflorum* and fermented *P. grandiflorum* extract, which was made with *Saccharomyces cerevisiae*, showed immunomodulatory effects in immune cells [15, 18]. Recently, a mixed extract of *P. grandiflorum*, *Apium graveolens*, and *Camellia sinensis* exhibited anti-obesity effects in an *in vivo* system [20]. Kang et al. [21] reported that a mixture of *Astragalus membranaceus*, *Schisandra chinensis*, and *P. grandiflorum* inhibited NO production by RAW264.7 cells.

*Salvia plebeia* R. Brown. belonging to the Lamiaceae family, is used as a medicinal plant in China, Korea, Japan, Afghanistan, India, Iran, and Australia [22, 23]. In Korea, *S. plebeian* has been used to treat the common cold, flu, and cough [22, 24]. *S. plebeia* contains chemical compounds that include flavonoids, lignans, phenolic compounds, terpenoids, sesquiterpenoids, diterpenoids, monoterpenoids, triterpenes, and volatile oils [22, 25–27], which showed biological activity, such as anti-tumor, antioxidant, hepatoprotective, antimicrobial, antiviral, and anti-inflammatory effects [24, 27–29]. *S. plebeian* can also stimulate the immune system by increasing phagocytic activity and NO, TNF-α, and IL-1β production in RAW264.7 cells, as well as promoting natural killer (NK) cell activity and IL-12 production in splenocytes [30]. In addition, a mixture of *S. plebeia* and Korean red ginseng exhibited anti-asthmatic and synergistic anti-inflammatory effects in airways [31–33].

Although individual immunostimulatory effects have been demonstrated, the mixture of *P. grandiflorum* and *S. plebeia* has not been previously studied, and the effects of the combination of these plants remain unclear. A number of studies demonstrated that the combination of plant materials enhanced immunity both *in vivo* and *in vitro*, showing a more powerful bioactive effect than single plant species alone [16, 34–37]. Therefore, the objective of this study was to determine if the effects on immune function are enhanced by treating murine macrophages with a mixture of *P. grandiflorum* and *S. plebeian* (PGSP).

## Materials and methods

### Preparation of PGSP

A mixture of PGSP was supplied after manufacturing by FD FARM Co.,Ltd. (Incheon, Korea). Briefly, dried *P. grandiflorum* and *S. plebeian* at a ratio of 1:1 (v/v) were mixed and extracted by adding distilled water at 100˚C ± 5 for 6–8 h. The water extract was concentrated using a vacuum concentrator until the solid content reached a brix of 50 ± 5.0 to obtain PGSP extract. PGSP was stored at -4˚C and used in this study. The ultra-performance liquid chromatography-tandem mass spectrometry (UPLC-MS/MS) to identify and quantify platycodin D and platycoside E in PSGP were presented in Supplemental Material. The UPLC-MS/MS analysis indicated that the PGSP contained 4 saponins, including platycoside D1 (1.11 ± 0.02 mg/g), platycoside D2 (1.73 ± 0.04 mg/g), platycoside E1 (2.40 ± 0.09 mg/g), and platycoside E2 (1.04 ± 0.09 mg/g).

## Cell culture and treatment

Mouse macrophage cell lines were obtained from the Korean Cell Line Bank (Korean Cell Line Research Foundation, Seoul, Korea) and maintained in RPMI-1640 medium supplemented with 10% fetal bovine serum (FBS, Welgene, Korea) and 1% penicillin/streptomycin (P/S, Welgene, Korea) in humidified incubators containing 5% $CO_2$. Various concentrations of PGSP were diluted in RPMI-1640 medium without phenol red, supplemented with 1% FBS and 1% P/S. Concentrations of 250, 500, 750, and 1000 μg/mL of PGSP were added to the cells for 24 h. Lipopolysaccharide (LPS, 1 μg/mL) was used as the positive control. After incubation, the immunological effects were assessed in subsequent experiments.

## Cell viability analysis

The EZ-Cytox Cell Viability Assay Kit (Daeil Labservice, Seoul, Korea) was used to determine cell viability. RAW264.7 cells ($1 \times 10^6$ cells/mL) were cultured with samples (250–1000 μg/mL) and incubated for 24 h. The supernatants were removed, and 100 μL of a water-soluble tetrazolium salt (WST) solution was added to each well. The PGSP-treated cells were incubated at 37˚C for 1 h. The absorbance was measured at 450 nm using a microplate reader (BioTek, Winooski, VT, USA).

## Determination of NO production

Griess reagent (Sigma-Aldrich, St. Louis, MO, USA) was used for the detection of NO production in macrophages. The cells ($1 \times 10^6$ cells/mL) were treated with PGSP for 24 h before the culture supernatants (100 μL) was transferred into each well of a 96-well plate. An equal volume (100 μL) of Griess reagent (0.1% $N$-1-napthylethylenediamine dihydrochloride in distilled water and 1% sulfanilamide in 5% phosphoric acid) was added to and incubated for 10 min at room temperature. Finally, the absorbance at 540 nm was measured using a microplate reader.

## Determination of PGE$_2$, COX-2, IL-1β, IL-6, and TNF-α production

Cells ($1 \times 10^6$ cells/mL) were treated with PGSP and subsequently cultured for 24 h. After treatment, the supernatants were collected and stored at -20˚C until analysis. PGE$_2$ levels were measured using a PGE$_2$ ELISA kit (Enzo Life Sciences, Farmingdale, NY, USA), and IL-1β, IL-6, TNF-α, and COX-2 concentrations were determined using ELISA kits specific for each cytokine (Abcam, Cambridge, UK), according to the manufacturer's instructions. The PGE$_2$, COX-2, IL-1β, IL-6, and TNF-α production were analyzed with each standard curve.

## Real-time qPCR analysis

Tri reagent® (Molecular Research Center, Inc., Cincinnati, OH, USA) was used to extract total RNA from LPS- and PGSP-treated and untreated RAW264.7 cells at a density of $1 \times 10^6$ cells/mL, according to the manufacturer's protocol. cDNA was synthesized using 1000 ng of RNA in the following steps: 25˚C for 10 min, 37˚C for 120 min, and 85˚C for 5 min, using the High-Capacity cDNA Reverse Transcription kit (Applied Biosystems, Foster City, CA, USA). To quantify the mRNA levels of gene expression, real-time qPCR analysis was performed using TB Green® Premix Ex Taq™ II (Takara Bio Inc., Shiga, Japan) that mixed a final concentration of 0.4 μM of a specific primer pairs, 1 × ROX Reference Dye, and 5 ng of cDNA templates, and then carried out on a QuantStudio™ 3 FlexReal-Time PCR System (Thermo Fisher Scientific, Waltham, MA, USA). The reaction steps were as follows: 95˚C for 30 s, 40 cycles of 95˚C for 5 s, and 60˚C for 34 s. Then, 95˚C for 30 s, 60˚C for 1 min, and 95˚C for 15 s were added at the end of the cycles. All reactions was carried out in triplicate. The relative

expression of the target genes was calculated and normalized to that of *β-actin* as an internal control. A list sequences of mouse primers are provided in Table 1.

## Assay of macrophage phagocytosis

The PGSP-treated cells at a density of $2 \times 10^6$ cells/mL were harvested and washed with ice-cold phosphate-buffered saline (PBS). After that, a solution of FITC-dextran (Sigma-Aldrich, USA) was added to PGSP-treated cells for 1 h at 37˚C. The reaction was stopped by adding 1 mL of ice-cold PBS, and the cells were washed three times with 1% paraformaldehyde. Phagocytic uptake was analyzed using the CytoFLEX Flow Cytometer (Beckman Coulter, Inc., USA).

## Analysis of western blotting

The PGSP-treated cells ($2 \times 10^6$ cells/mL) were extracted in 120 μL of lysis buffer containing radioimmunoprecipitation assay (RIPA) buffer, 0.5 mM ethylenediaminetetraacetic acid (EDTA), and an inhibitor cocktail (protease and phosphatase). The cell lysates were incubated at 4˚C for 30 min and then centrifuged at 13,000 rpm for 20 min. The protein concentration in the supernatant was determined using the Pierce™ BCA Protein Assay Kit (Thermo Fisher Scientific, USA) and estimated using the standard curve, as directed by the manufacturer. Subsequently, equal volumes of proteins were resolved on sodium dodecyl sulfate-polyacrylamide gels and then transferred to polyvinylidene membranes. The membranes were blocked for 1 h in 5% skim milk in $1 \times$ Tris-buffered saline containing 1% Tween-20 ($1 \times$ TBST). After incubation, the membranes were washed with $1 \times$ TBST and incubated overnight with the primary antibody (1:2000). Then, the membranes were washed and incubated for 1 h with horseradish peroxidase-conjugated goat anti-rabbit IgG (H+L) (1:2000). The protein bands were detected by Pierce® ECL Plus Western Blotting Substrate and visualized with the ChemiDoc XRS + imaging system (Bio-Rad, Hercules, CA, USA).

## Data analysis

The data were analyzed using SPSS version 23.0 software (SPSS, Inc., Chicago, IL, USA) and presented as the mean ± standard deviation (SD). Each value is the mean of three separate experiments ($n = 3$). The significance was analyzed using a one-way analysis of variance followed by Duncan's new multiple range test. $P$-values of $< 0.05$ were considered statistically significant.

**Table 1. Primer sequences used in real-time qPCR.**

| Target genes | Accession numbers | Primer sequence (5′ to 3′) | |
|---|---|---|---|
| | | **Sense** | **Antisense** |
| IL-1β | NM_008361.4 | GGGCCTCAAAGGAAAGAATC | TACCAGTTGGGGAACTCTGC |
| IL-4 | NM_021283.2 | ACAGGAGAAGGGACGCCAT | GAAGCCCTACAGACGAGCTCA |
| IL-6 | NM_031168.2 | AGTTGCCTTCTTGGGACTGA | CAGAATTGCCATTGCACAAC |
| TNF-α | D84199.2 | ATGAGCACAGAAAGCATGATC | TACAGGCTTGTCACTCGAATT |
| iNOS | BC062378.1 | TTCCAGAATCCCTGGACAAG | TGGTCAAACTCTTGGGGTTC |
| COX-2 | NM_011198.4 | AGAAGGAAATGGCTGCAGAA | GCTCGGCTTCCAGTATTGAG |
| β-actin | NM_007393.5 | CCACAGCTGAGAGGGAAATC | AAGGAAGGCTGGAAAAGAGC |

## Results

### Effect of PGSP on cell viability

As shown in Fig 1, the cell viability after treatment with PGSP concentrations of 250, 500, 750, and 1000μg/mL was 101.12 ± 1.35%, 98.07 ± 0.59%, 98.48 ± 0.71%, and 100.59 ± 0.77%, respectively, but statistically different from the RPMI controls ($p < 0.05$). Neither PGSP nor LPS were cytotoxic to RAW264.7 cells within the tested concentration range, suggesting that these doses of PGSP could be used in further experiments without causing cytotoxicity.

### Effects of PGSP on the NO production

After 24 h of incubation with PGSP, the production of NO in RAW264.7 macrophages was measured. As shown in Fig 2A, PGSP showed a significant stimulatory effect on NO production in a dose-dependent manner, which was 81.19 ± 1.30% at the highest concentration (1000 μg/mL). The amount of NO produced by PGSP was comparable to that of the positive control, LPS (1 μg/mL), indicating that it exerted strong stimulant activity.

### Effects of PGSP on the production of PGE₂, COX-2, IL-1β, IL-6, and TNF-α

Cells treated with PGSP (250–1000 μg/mL) or LPS (1 μg/mL) secreted more PGE$_2$, COX-2, IL-1β, IL-6, and TNF-α than cells treated with RPMI (Fig 2B–2F). The amounts of cytokines secreted from PGSP-treated cells were lower than those from LPS-treated cells (the positive control). The production of PGE$_2$, COX-2, IL-1β, IL-6, and TNF-α was also increased by 88.05 ± 0.64%, 73.17 ± 4.06%, 77.30 ± 0.79%, 100.65 ± 0.52%, and 89.51 ± 1.50%, respectively, at 1000 μg/mL of PGSP.

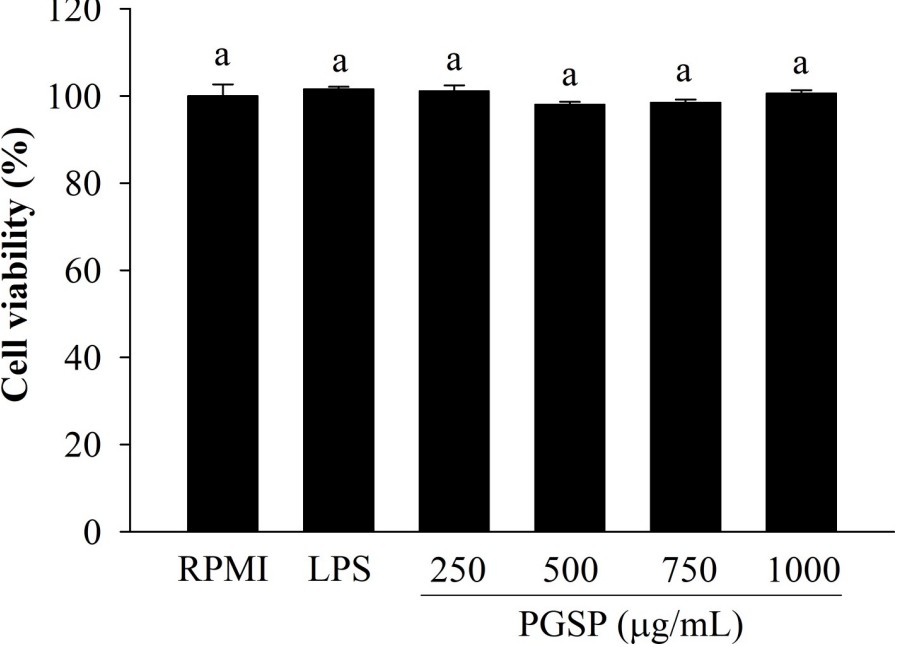

**Fig 1. Effect of PGSP on the viability of RAW264.7 macrophages.** Cell viability was determined by the EZ-Cytox Cell Viability Assay Kit. All values are presented as the mean ± SD of three independent experiments ($n = 3$). A different letter ($p < 0.05$) reveals statistically significant differences within treatments.

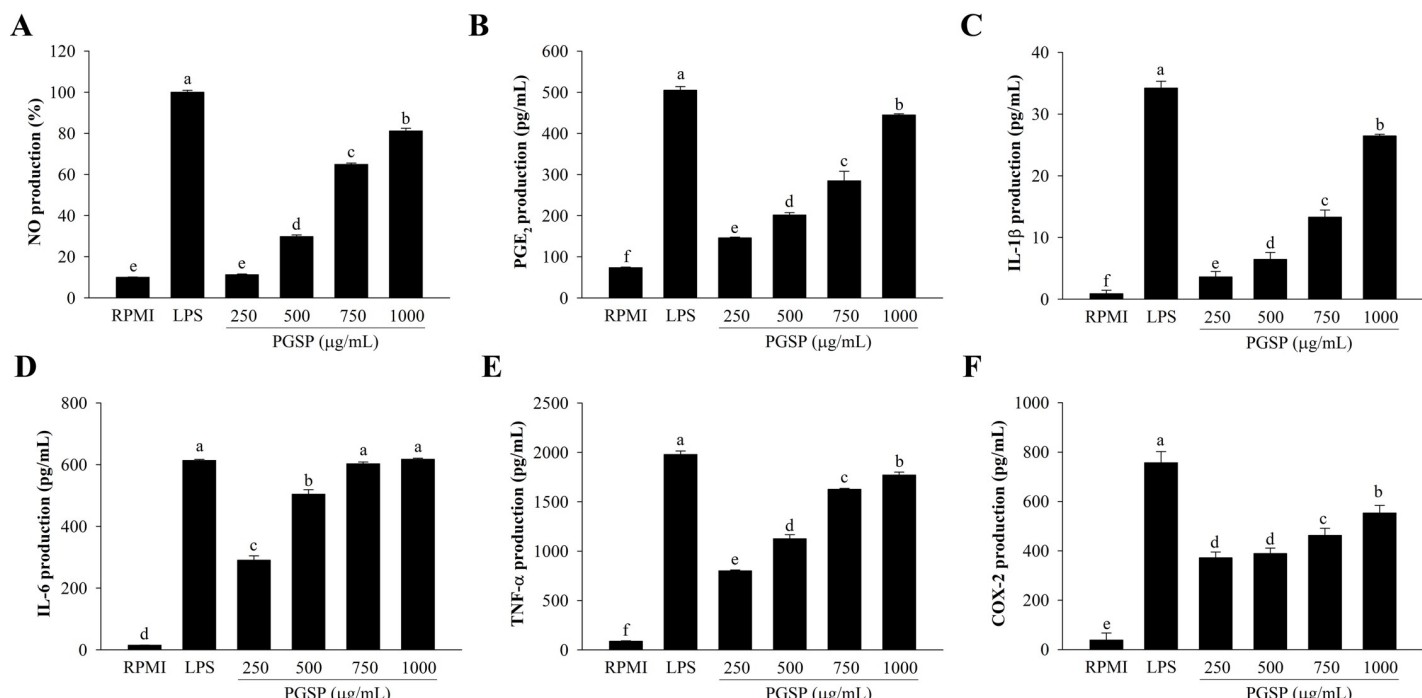

**Fig 2. Effect of PGSP on inflammatory mediator production in RAW264.7 macrophages.** (A) NO concentration in the medium was measured using Griess reagent. (B–D) $PGE_2$, COX-2, IL-1β, IL-6, and TNF-α production was determined using an ELISA kit. All values are presented as the mean ± SD of three independent experiments ($n = 3$). A different letter ($p < 0.05$) reveals statistically significant differences within treatments.

### Effects of PGSP on the cytokine expression

Real-time qPCR analysis showed that PGSP increased the mRNA levels of *IL-1β*, *IL-4*, *IL-6*, and *TNF-α* in RAW264.7 macrophages (Fig 3). Furthermore, the immune-stimulatory mRNA levels of cytokines, such as *IL-1β*, *IL-4*, *IL-6*, and *TNF-α*, were higher in macrophage cells treated with PGSP compared to RPMI-treated cells.

### Effect of PGSP on iNOS and COX-2 mRNA and protein expression

Real-time qPCR and western blot analyses were performed on RAW264.7 macrophages to examine whether the induction of immunomodulators like NO and $PGE_2$ occurred due to mRNA and protein expression of iNOS and COX-2. As shown in Fig 4A and 4B, the mRNA expression levels of *iNOS* and *COX-2* were significantly enhanced in a concentration-dependent manner after treatment of the cells with PGSP for 24 h. PGSP concentration-dependently induced the expression of iNOS and COX-2 protein (Fig 4C). Additionally, PGSP induced iNOS and COX-2 expression at a lower level than that in LPS-treated cells but stronger than that in RPMI-treated cells. Therefore, PGSP may regulate pro-inflammatory mediators, such as iNOS and COX-2.

### Effects of PGSP on phagocytic activity

A fluorescein isothiocyanate (FITC)–dextran uptake assay was performed to measure the effects of PGSP or LPS on macrophage phagocytosis. As shown in Fig 5, the results revealed that LPS could increase macrophage phagocytic activity by 68.77 ± 0.07%, which was displayed by higher FITC-fluorescence intensity than the untreated control group. Flow cytometry revealed that the percentage of phagocytosis increased to 41.02 ± 2.27%, 54.69 ± 2.70%,

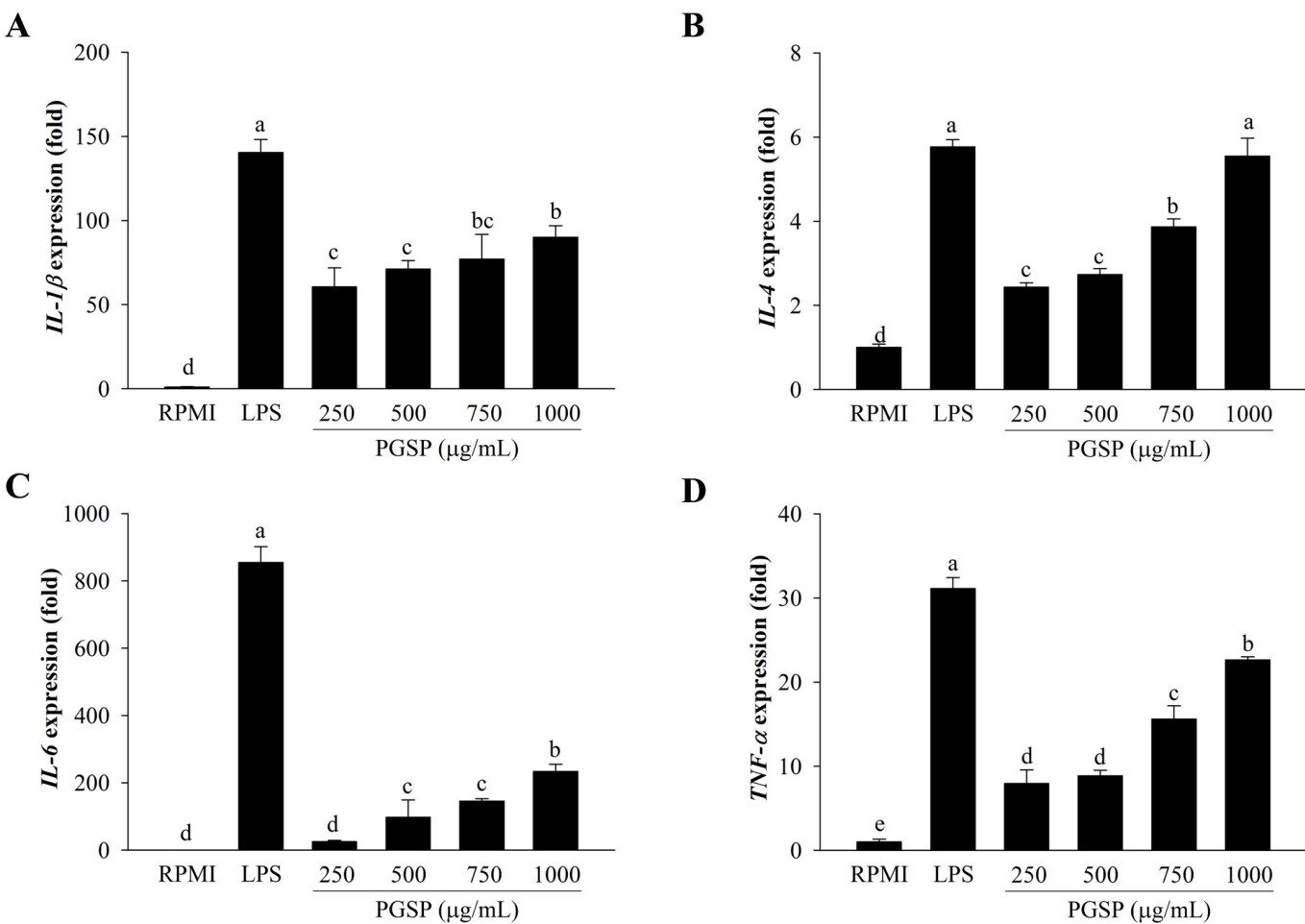

**Fig 3. Effect of PGSP on cytokine expression in RAW264.7 macrophages.** The mRNA expression of cytokines was determined by real-time qPCR. All values are presented as the mean ± SD of three independent experiments ($n$ = 3). A different letter ($p$ < 0.05) reveals statistically significant differences within treatments.

62.72 ± 2.90%, and 69.22 ± 1.59% in cells treated with 250, 500, 750, and 1000 μg/mL, respectively.

## Effects of PGSP on NF-κB and MAPK activation

To investigate the molecular mechanisms underlying the immune-enhancing effects of PGSP, the phosphorylation of NF-κB subunit p65, JNK, ERK, and p-38 MAPK was evaluated in RAW264.7 cells by western blotting, and the results are shown in Fig 6. A dose-dependent increase in NF-κB and MAPK activation was observed by treatment with PGSP compared to RPMI. PGSP induced NF-κB-p65, ERK1/2, p38, and JNK phosphorylation, indicating NF-κB and MAPK activation. LPS also strongly increased NF-κB and MAPK expression levels (Fig 6).

## Discussion

Previous pharmacological studies on *P. grandiflorum* mainly examined its immune-enhancing effects *in vitro* and *in vivo* [15, 18]. *P. grandiflorum* extracts primarily contained platycodin D and exhibited immuno-enhancing effects both *in vitro* and *in vivo* [16, 17, 38]. Treatment of RAW264.7 cells with the combination of fermented *P. grandiflorum* and soybean extract or

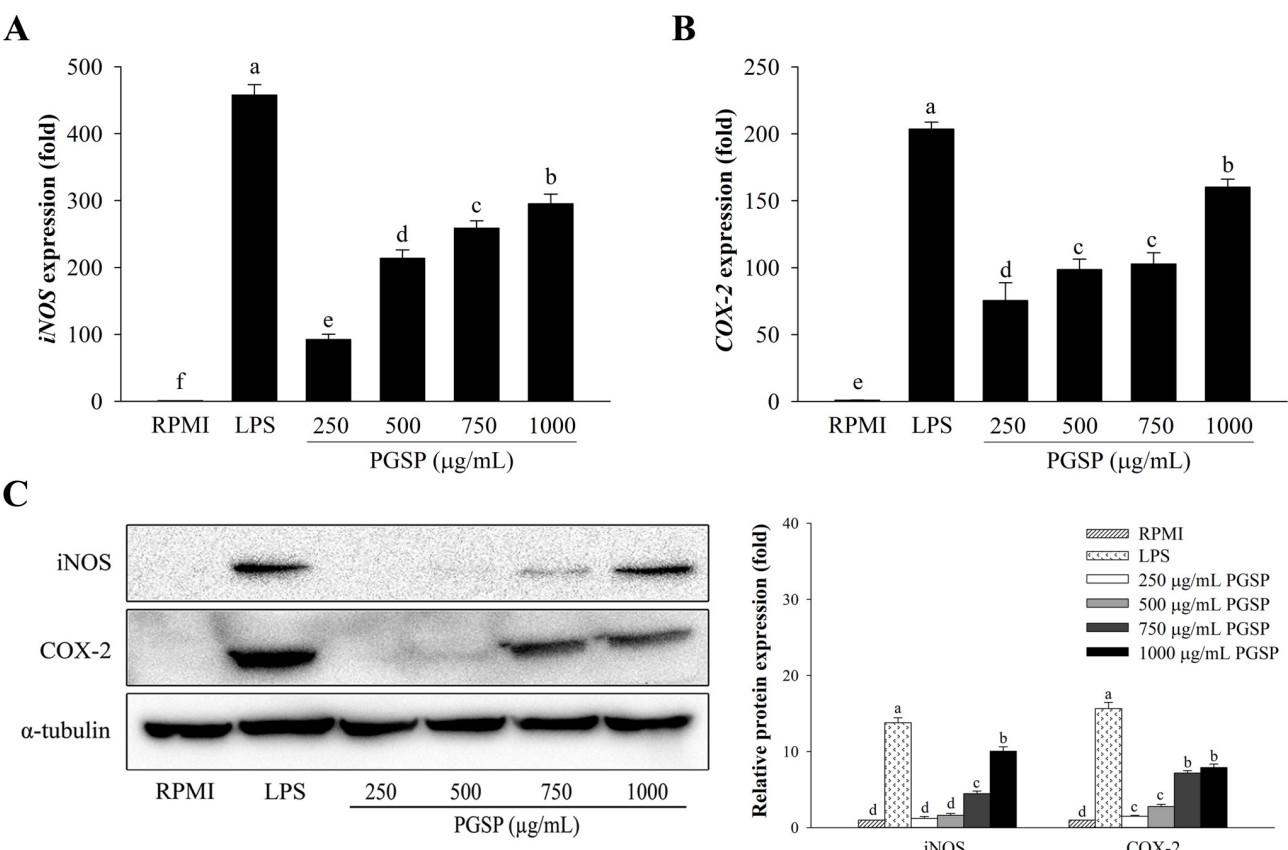

**Fig 4. Effect of PGSP on cytokine expression in RAW264.7 macrophages.** The mRNA expression of *iNOS* (A) and *COX-2* (B) was determined by real-time qPCR. (C) The protein expression of iNOS and COX-2 was determined by western blotting. All values are presented as the mean ± SD of three independent experiments ($n = 3$). A different letter ($p < 0.05$) reveals statistically significant differences within treatments.

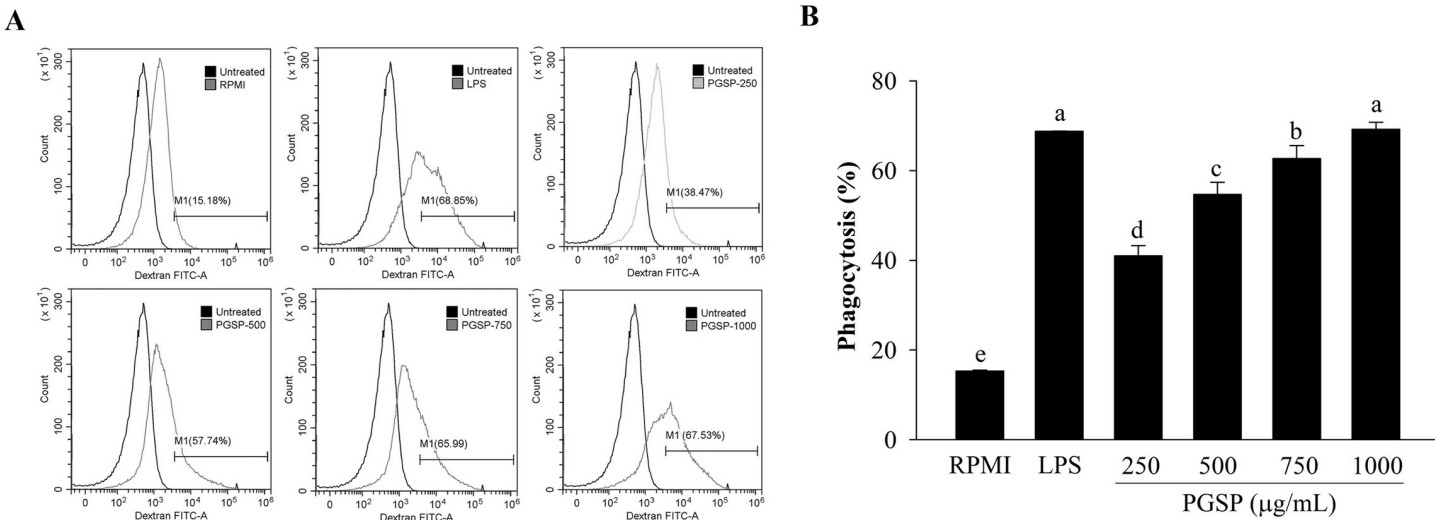

**Fig 5. Effect of PGSP on the phagocytic ability of macrophages.** The cellular uptake of FITC-dextran was determined by flow cytometry. (A) Flow data for the uptake by FITC-dextran of macrophages from one of three independent experiments. (B) The proportion of phagocytic macrophage activity. All values are presented as the mean ± SD of three independent experiments ($n = 3$). A different letter ($p < 0.05$) reveals statistically significant differences within treatments.

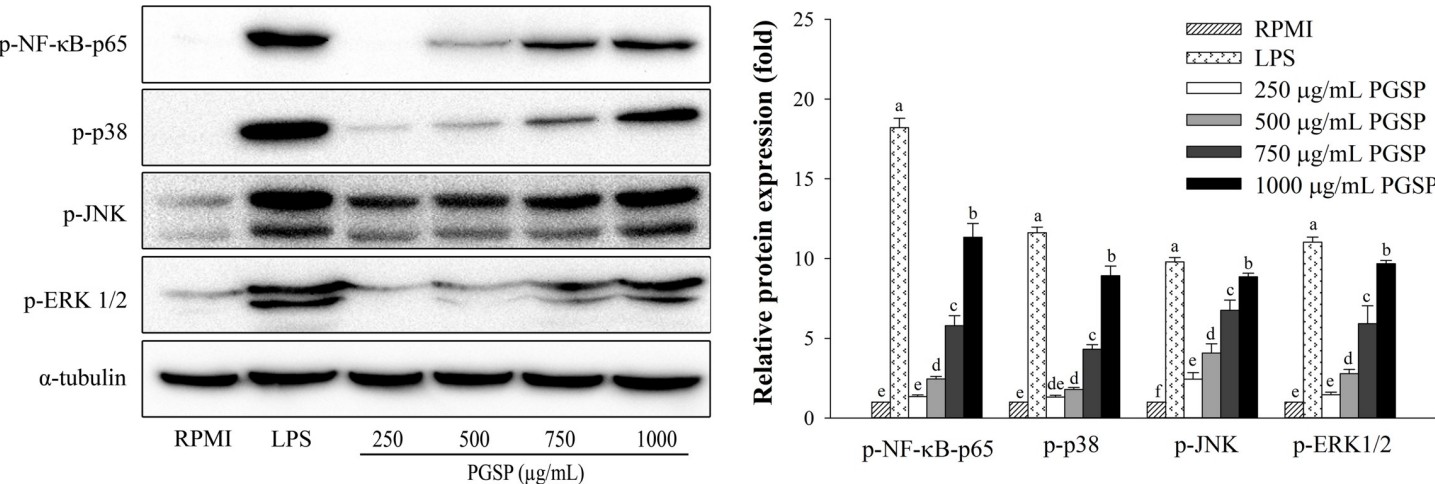

**Fig 6. Effects of PGSP on NF-κB and MAPK activation in RAW264.7 macrophages.** Phosphorylation levels of ERK, JNK, and p38 MAPKs were determined by western blotting. All values are presented as the mean ± SD of three independent experiments ($n = 3$). A different letter ($p < 0.05$) reveals statistically significant differences within treatments.

vitamin C showed immune responses greater than treatment with fermented *P. grandiflorum* alone [16]. *S. plebeian* extracts were also used in these studies to modulate immunity in the forced swimming exercise-induced model [30]. *S. plebeian* extracts contained higher quantities of flavonoids, such as luteolin, luteoloside, nepetin, nepitrin, hispidulin, homoplantagenin, and eupatorine [39], which have anti-inflammatory effects. These flavonoid compounds displayed immunomodulatory effects in various cells [40]. However, the action of *P. grandiflorum* mixed with *S. plebeian* has not been reported. In this study, we investigated the immunostimulatory effects of a mixture of *P. grandiflorum* and *S. plebeia* (PGSP) in a macrophage cell line and the mechanism through which PGSP enhanced inflammatory responses.

Macrophages are known to be involved in multiple biological functions, such as inflammation control and immune enhancement, by releasing pro-inflammatory cytokines and inflammatory factors [41, 42]. The phagocytic activity of macrophages is a key indicator of macrophage activation in non-specific immune responses [43]. The results of this study indicated that PGSP significantly improved macrophage phagocytosis, consistent with previously reported results [43–46]. The results showed that PGSP activated macrophages to improve immunity.

The signal transduction molecules NO and PGE$_2$ play important roles in inflammation. Fermented *P. grandiflorum* extract enhanced NO production at the highest concentration (500 μg/mL), but the effect was only slightly statistically significant compared to the control group [15]. In this study, PGSP significantly stimulated NO production at a dose range of 250–1000 μg/mL. Similarly, mixtures of *Sasa quelpaertensis* Nakai and *Ficus erecta* var. *sieboldin* effectively stimulated NO production by RAW264.7 cells in a dose range of 10–1000 μg/mL [37]. The inflammatory response involves the enzymes iNOS and COX-2, which regulate the production of NO and PGE$_2$. The results of this study showed that PGSP increased NO and PGE$_2$ secretion and upregulated *iNOS* and *COX-2* expression. PGSP also enhanced immunity by upregulating iNOS and COX-2 protein expression in macrophage cells. Ko et al. (2012) [1] investigated the immunomodulatory effects of *Abelmoschus esculentus* extracts on RAW2647 cells by regulating the production of TNF-α, IL-1β, IL-6, NO, and PGE$_2$, as well as the expression of iNOS and COX-2 [1]. The combination of *Panax ginseng* and *Scrophularia*

*buergeriana* increased phagocytosis, cytokine, and NO production and enhanced *iNOS* expression via NF-κB activation in RAW264.7 cells and immune responses in splenocytes [46].

Cytokines are important mediators involved in modulating immune and inflammatory responses [42]. A variety of cytokines, including TNF-α, IL-1β, and IL-6, are potent immuno-modulators in activated macrophages [47, 48]. According to our findings, PGSP increased serum IL-1β, IL-6, and TNF-α levels. PGSP also increased the mRNA expression levels of cytokines, such as *IL-1β*, *IL-4*, *IL-6*, and *TNF-α*, in RAW264.7 macrophages. Many studies showed that plant extracts enhanced immunity by upregulating enzymes (iNOS and COX-2) and cytokines (TNF-α, IL-1β, IL-6, IL-10, and IL-12) in macrophages [2, 44, 47]. Thus, our results suggested that PGSP improved immune function by increasing immunostimulatory cytokines, as previous reports showed.

Macrophage activation is associated with the generation of immunomodulators through MAPK and NF-κB pathways [1, 2]. *P. grandiflorum* polysaccharides were shown to activate MAPK and AP-1 in macrophages [49]. Our study also demonstrated that PGSP could dose-dependently induce the activation of signaling molecules, such as NF-κB and MAPKs, in RAW264.7 cells, indicating that these molecules regulated a variety of cellular functions, including cell survival, proliferation, and apoptosis [50]. These results suggested that PGSP strongly enhanced macrophage cellular immunity by stimulating the phosphorylation of ERK, JNK, p38 MAPK, and NF-κB p65. *Althaea rosea* extracts were found to exhibit immunomodu-latory activity by increasing NO, cytokines, and mediator secretion by activating NF-κB and MAPK proteins in macrophages [48]. Therefore, the present results suggest that PGSP enhanced immune responses by stimulating macrophages through NF-κB and MAPK signaling.

## Conclusion

PGSP showed strong immunoenhancing activity in macrophages by increasing the secretion of NO, PGE$_2$, cytokines, mediators, and phagocytosis, as well as increasing the expression of immune-related genes and the activation of NF-κB and MAPK signaling pathways. Thus, PGSP could act as a potent immunomodulator derived from *P. grandiflorum* and *S. plebeian* and could also be developed as a functional food or supplementary diet ingredient for health.

## Supporting information

**S1 Fig. Original western blot gel image data.**
(PDF)

**S1 File. Supplementary materials and methods.**
(PDF)

## Author Contributions

**Conceptualization:** Woo Jung Park.

**Data curation:** Weerawan Rod-in, Woo Jung Park.

**Formal analysis:** A-yeong Jang, Minji Kim.

**Funding acquisition:** Woo Jung Park.

**Investigation:** A-yeong Jang, Minji Kim.

**Methodology:** A-yeong Jang, Minji Kim, Weerawan Rod-in.

**Project administration:** Woo Jung Park.

**Resources:** Woo Jung Park.

**Software:** A-yeong Jang, Minji Kim, Weerawan Rod-in.

**Supervision:** Woo Jung Park.

**Validation:** A-yeong Jang, Minji Kim.

**Visualization:** A-yeong Jang, Minji Kim, Weerawan Rod-in.

**Writing – original draft:** A-yeong Jang.

**Writing – review & editing:** Yu Suk Nam, Tae Young Yoo.

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
