## [Decision Letter · Decision Letter 0]

17 Nov 2023

PONE-D-23-29766In vitro immune-enhancing effects of Platycodon grandiflorumcombined with Salvia plebeian via MAPK and NF-κB signaling in RAW264.7 cellsPLOS ONE

Dear Dr. Park,

Thank you for submitting your manuscript to PLOS ONE. After careful consideration, we feel that it has merit but does not fully meet PLOS ONE’s publication criteria as it currently stands. Therefore, we invite you to submit a revised version of the manuscript that addresses the points raised during the review process.

We look forward to receiving your revised manuscript.

Kind regards,

Babatunde Olanrewaju Motayo, Ph.D.

Academic Editor

PLOS ONE

Journal Requirements:

-https://doi.org/10.1016/j.jff.2020.104139

4. In your revision ensure you cite all your sources (including your own works), and quote or rephrase any duplicated text outside the methods section. Further consideration is dependent on these concerns being addressed.

6. Thank you for stating the following financial disclosure: 

NO - Include this sentence at the end of your statement: The funders had no role in study design, data collection and analysis, decision to publish, or preparation of the manuscript.

7. In your Data Availability statement, you have not specified where the minimal data set underlying the results described in your manuscript can be found. PLOS defines a study's minimal data set as the underlying data used to reach the conclusions drawn in the manuscript and any additional data required to replicate the reported study findings in their entirety. All PLOS journals require that the minimal data set be made fully available. For more information about our data policy, please see http://journals.plos.org/plosone/s/data-availability.

8. PLOS ONE now requires that authors provide the original uncropped and unadjusted images underlying all blot or gel results reported in a submission’s figures or Supporting Information files. This policy and the journal’s other requirements for blot/gel reporting and figure preparation are described in detail at https://journals.plos.org/plosone/s/figures#loc-blot-and-gel-reporting-requirements and https://journals.plos.org/plosone/s/figures#loc-preparing-figures-from-image-files. When you submit your revised manuscript, please ensure that your figures adhere fully to these guidelines and provide the original underlying images for all blot or gel data reported in your submission. See the following link for instructions on providing the original image data: https://journals.plos.org/plosone/s/figures#loc-original-images-for-blots-and-gels. 

Reviewers' comments:

Reviewer's Responses to Questions

**Comments to the Author**

1. Is the manuscript technically sound, and do the data support the conclusions?

Reviewer #1: Yes

Reviewer #2: Partly

2. Has the statistical analysis been performed appropriately and rigorously? 

Reviewer #1: Yes

Reviewer #2: N/A

3. Have the authors made all data underlying the findings in their manuscript fully available?

Reviewer #1: Yes

Reviewer #2: Yes

4. Is the manuscript presented in an intelligible fashion and written in standard English?

Reviewer #1: Yes

Reviewer #2: Yes

5. Review Comments to the Author

Reviewer #1: GENERAL REVIEW

A very beautiful work. But more explanation is required under the methodology. A precise description of the experimentations will be more beneficial to readers. Results should also be presented in areas that they were expressed in figures in the way they were described under methodology (mean ± SD).

SPECIFIC REVIEWS

Line 43: all enhance – all, before enhance

Line 96 – A good description of the laboratory procedure till how the cell culture supernatants were harvested should be described.

Lines 97 – 101 : Cell viability analysis

A step-by-step-description of how this was carried out will be more beneficial to the readers. For instance, “Water-soluble tetrazolium salt (WST) solution was added to each well and incubated for 1 h”. – Which well was it added to? What did the well contain? All the steps should be well described.

Lines 102 – 106 : Determination of NO Production

The description of the methodology is not explicit enough. It will be beneficial if the laboratory analysis is well summarized. What is the volume of Greiss reagent added to what volume culture supernatants before incubation and measurement?

Line 109 – Should have come as the last lines under the “Cell culture” [line 87] heading. (See my comment on line 96 above).

Line 110-112: Please describe briefly the principle of the ELISA and describe briefly the step-by-step analytical method employed in carrying out the technique. It can be generalized for all the ELISA assays.

Line 123 – Please state in your Real-time qPCR analysis, where the primers were sourced from?

Line 132 – What is RIPA buffer? Please write in full, the bracket the abbreviation

Line 133 – EDTA, as in 132.

Lines 134 – 136: briefly describe how you measured the protein. Not just to write that they were measured

Lines 146-147 – “The significance was analyzed using one-way analysis of variance” – this is not a correct statement. Analysis of variance is used when testing more than 2 parametric variables, but this is not the right way to express it. Please correct.

Lines 151 – 152: Your n=3, and express=ion of the results were in box plot. Therefore, you should express your figurative reports in XXX ± XX, (mean ± SD) and not just in percentage.

Lines 178 – 179 – Express your results in mean ± SD as explained above

Lines 183 – 184 – Express in figures like above.

Lines 206 – 212 – You need to express your results generally in mean ± SD as you express in your methodology, and as the figures show.

Reviewer #2: The authors investigated the immune enhancing effects of PGSP via MAPK and NF-KB signaling in RAW264.7 cells. They related the genetic expression (mRNA) and protein levels of inflammatory markers (COX-2, TNF-alpha, IL-1B, IL-6, iNOS, etc) as basis for immune enhancing effects. They also attempted to describe the mechanisms of stimulation as being through phosphorylation of ERK,JNK,p38 and NF-kBp65.

However, there are major concerns to the method use in this study.

1. The LPS control is at a concentration of 1ug/ml while the PGSP extracts are at 250-1000ug/ml. The results show that at 250ug, the levels of inflammatory markers (iNOS and COx2) on WB is almost nil (see figure 4c). PGSP extracts do not seem to stimulate inflammatory markers at low doses. Comparing their data with a control at a low dose is not a very good basis for comparison. The authors did not also show the levels of cytotoxicty due to this high dose extract except cell counts for cell viability. There are inflammatory markers of cell death and apoptosis (eg PD1). It would be good to show the levels of such markers in the study.

2. The authors claim that there are synergistic effects due to the two combined extracts. However, they fail to show data for levels of inflammatory markers for each of the extracts.

Minor Concerns

1. Why was the extract diluted in 1% FBS and P/S and not in plain medium?

2.The authors need to state the concentration of the cells used for RNA extraction, Western blot as well as Macrophage phagocytosis assay.

3. The authors did not state clearly how the mRNA expression analysis was done. They seem to use Actin as control but they did not say how many biological and technical replicates were done and how they analysed for mRNA expression.

6. PLOS authors have the option to publish the peer review history of their article (what does this mean?). If published, this will include your full peer review and any attached files.

Reviewer #1: **Yes: **Dr. Kazeem S. Akinwande

Reviewer #2: No

---

## [Author Response · Author response to Decision Letter 0]

15 Dec 2023

Response to Reviewer 1 Comments

GENERAL REVIEW

A very beautiful work. But more explanation is required under the methodology. A precise description of the experimentations will be more beneficial to readers. Results should also be presented in areas that they were expressed in figures in the way they were described under methodology (mean ± SD).

- It was corrected.

- Thanks.

SPECIFIC REVIEWS

Line 43: all enhance – all, before enhance

- It was corrected.

- Thanks.

Line 96 – A good description of the laboratory procedure till how the cell culture supernatants were harvested should be described.

- It was corrected.

- Thanks.

Lines 97 – 101 : Cell viability analysis

A step-by-step-description of how this was carried out will be more beneficial to the readers. For instance, “Water-soluble tetrazolium salt (WST) solution was added to each well and incubated for 1 h”. – Which well was it added to? What did the well contain? All the steps should be well described.

- It was corrected.

- Thanks.

Lines 102 – 106 : Determination of NO Production

The description of the methodology is not explicit enough. It will be beneficial if the laboratory analysis is well summarized. What is the volume of Greiss reagent added to what volume culture supernatants before incubation and measurement?

- It was corrected.

- Thanks.

Line 109 – Should have come as the last lines under the “Cell culture” [line 87] heading. (See my comment on line 96 above).

Line 110-112: Please describe briefly the principle of the ELISA and describe briefly the step-by-step analytical method employed in carrying out the technique. It can be generalized for all the ELISA assays.

- It was corrected.

- Thanks.

Line 123 – Please state in your Real-time qPCR analysis, where the primers were sourced from?

- All primers were based on mouse (Mus musculus) transcript sequence. 

- Thank you.

Line 132 – What is RIPA buffer? Please write in full, the bracket the abbreviation

- It was corrected.

- Thanks.

Line 133 – EDTA, as in 132.

- It was corrected.

- Thanks.

Lines 134 – 136: briefly describe how you measured the protein. Not just to write that they were measured

- It was corrected.

- Thanks.

Lines 146-147 – “The significance was analyzed using one-way analysis of variance” – this is not a correct statement. Analysis of variance is used when testing more than 2 parametric variables, but this is not the right way to express it. Please correct.

- It was corrected.

- Thanks.

Lines 151 – 152: Your n=3, and expression of the results were in box plot. 

Therefore, you should express your figurative reports in XXX ± XX, (mean ± SD) and not just in percentage.

- It was corrected.

- Thanks.

Lines 178 – 179 – Express your results in mean ± SD as explained above

- It was corrected.

- Thanks.

Lines 183 – 184 – Express in figures like above.

- It was corrected.

- Thanks.

Lines 206 – 212 – You need to express your results generally in mean ± SD as you express in your methodology, and as the figures show.

- It was corrected.

- Thanks.

Response to Reviewer 2 Comments

Reviewer #2: The authors investigated the immune enhancing effects of PGSP via MAPK and NF-KB signaling in RAW264.7 cells. They related the genetic expression (mRNA) and protein levels of inflammatory markers (COX-2, TNF-alpha, IL-1B, IL-6, iNOS, etc) as basis for immune enhancing effects. They also attempted to describe the mechanisms of stimulation as being through phosphorylation of ERK, JNK, p38 and NF-kBp65. However, there are major concerns to the method use in this study. 

1. The LPS control is at a concentration of 1ug/ml while the PGSP extracts are at 250-1000ug/ml. The results show that at 250ug, the levels of inflammatory markers (iNOS and COx2) on WB is almost nil (see figure 4c). PGSP extracts do not seem to stimulate inflammatory markers at low doses. Comparing their data with a control at a low dose is not a very good basis for comparison. 

- LPS has been used as a positive control to stimulate the macrophage cells, and this concentration has been used for immune response research. 

- Many previous researches like the followings have shown to use this concentration even though the concentration between the control group and samples is different. 

References

• Zhou, Y., Qian, C., Yang, D., Tang, C., Xu, X., Liu, E. H., ... & Zhao, Z. (2021). Purification, structural characterization and immunomodulatory effects of polysaccharides from Amomum villosum Lour. on RAW 264.7 macrophages. Molecules, 26(9), 2672.

• Lin, S., Li, H. Y., Yuan, Q., Nie, X. R., Zhou, J., Wei, S. Y., ... & Wu, D. T. (2020). Structural characterization, antioxidant activity, and immunomodulatory activity of non-starch polysaccharides from Chuanminshen violaceum collected from different regions. International Journal of Biological Macromolecules, 143, 902-912.

• Tran, T. H. M., Mi, X. J., Huh, J. E., Mitra, P. A., & Kim, Y. J. (2023). Cirsium japonicum var. maackii fermented with Pediococcus pentosaceus induces immunostimulatory activity in RAW 264.7 cells, splenocytes and CTX-immunosuppressed mice. Journal of Functional Foods, 102, 105449.

• Jiang, S., Yin, H., Qi, X., Song, W., Shi, W., Mou, J., & Yang, J. (2021). Immunomodulatory effects of fucosylated chondroitin sulfate from Stichopus chloronotus on RAW 264.7 cells. Carbohydrate Polymers, 251, 117088.

- In addition, the results were corrected by comparing the significant differences within all treatments even though the signal is not strong and other previous reports have shown like our data.

References

• Lee, J., Kim, S., & Kang, C. H. (2022). Immunostimulatory activity of lactic acid bacteria cell-free supernatants through the activation of NF-κB and MAPK signaling pathways in RAW 264.7 cells. Microorganisms, 10(11), 2247.

• Jung, J. I., Lee, H. S., Kim, S. M., Kim, S., Lim, J., Woo, M., & Kim, E. J. (2022). Immunostimulatory activity of hydrolyzed and fermented Platycodon grandiflorum extract occurs via the MAPK and NF-κB signaling pathway in RAW 264.7 cells. Nutrition Research and Practice, 16(6), 685-699.

• Kim, J. S., Lee, E. B., Choi, J. H., Jung, J., Jeong, U. Y., Bae, U. J., ... & Lee, S. H. (2023). Antioxidant and immune stimulating effects of Allium cepa skin in the raw 264.7 cells and in the C57BL/6 mouse immunosuppressed by cyclophosphamide. Antioxidants, 12(4), 892.

• Gao, X., Qi, J., Ho, C. T., Li, B., Mu, J., Zhang, Y., ... & Xie, Y. (2020). Structural characterization and immunomodulatory activity of a water-soluble polysaccharide from Ganoderma leucocontextum fruiting bodies. Carbohydrate polymers, 249, 116874.

• Cho, H. Y., Lee, J. E., Lee, J. H., Ahn, D. U., & Paik, H. D. (2023). The immune-enhancing activity of egg white ovalbumin hydrolysate prepared with papain via MAPK signaling pathway in RAW 264.7 macrophages. Journal of Functional Foods, 103, 105487.

- Thanks.

The authors did not also show the levels of cytotoxicity due to this high dose extract except cell counts for cell viability. There are inflammatory markers of cell death and apoptosis (eg PD1). It would be good to show the levels of such markers in the study.

- Thank you very much for your valuable comment and suggestion.

- In this current study, the purpose of the experiment was focused on testing the percentage of cell viability and finding the concentration of the extract that does not cause cell death and is effective in increasing immunity.

- Additionally, previous many references such as the followings have shown the cytotoxicity similar to ours.

References

• Eo, H. J., Shin, H., Song, J. H., & Park, G. H. (2021). Immuno-enhancing effects of fruit of Actinidia polygama in macrophages. Food and Agricultural Immunology, 32(1), 754-765.

• Tran, T. H. M., Mi, X. J., Huh, J. E., Mitra, P. A., & Kim, Y. J. (2023). Cirsium japonicum var. maackii fermented with Pediococcus pentosaceus induces immunostimulatory activity in RAW 264.7 cells, splenocytes and CTX-immunosuppressed mice. Journal of Functional Foods, 102, 105449.

• Dong, Z., Zhang, M., Li, H., Zhan, Q., Lai, F., & Wu, H. (2020). Structural characterization and immunomodulatory activity of a novel polysaccharide from Pueraria lobata (Willd.) Ohwi root. International journal of biological macromolecules, 154, 1556-1564.

• He, P., Pan, L., Wu, H., Zhang, L., Zhang, Y., Zhang, Y., ... & Zhang, M. (2022). Isolation, identification, and immunomodulatory mechanism of peptides from Lepidium meyenii (Maca) protein hydrolysate. Journal of Agricultural and Food Chemistry, 70(14), 4328-4341.

• Park, C., HwangBo, H., Lee, H., Kim, G. Y., Cha, H. J., Choi, S. H., ... & Choi, Y. H. (2020). The immunostimulatory effect of indole-6-carboxaldehyde isolated from Sargassum thunbergii (Mertens) Kuntze in RAW 264.7 macrophages. Animal Cells and Systems, 24(4), 233-241.

2. The authors claim that there are synergistic effects due to the two combined extracts. However, they fail to show data for levels of inflammatory markers for each of the extracts.

- In our study, the effect of Platycodon grandiflorum alone on NO production at the concentrations of 250-1000 μg/ml for pre-experiments (data not shown in the manuscript) were measured.

- Compared with LPS (100%), the treatment of P. grandiflorum alone has slightly increased NO production to be 8.05-8.96% (A), while the mixture of P. grandiflorum and S. plebeian (PGSP) showed NO production to be 11.27-81.19% (B), that data showed in manuscript. Therefore, the results indicated that the PGSP were high efficiency more than P. grandiflorum alone.

- In addition, many previous studies reported that the immune-enhancement activity of only mixture of various plants have been shown and do not perform the experiments separately with individual substances.

- Thanks.

References

• Choi, M. (2022). Immunity-enhancing effect of extracts extracted from leaves of Rubia hexaphylla, Cymbopogon citratus, and Dioscorea japonica for sustainable healthy life. Sustainability 2022, 14, 2804.

• Han, N. R., Kim, K. C., Kim, J. S., Ko, S. G., Park, H. J., & Moon, P. D. (2022). The immune-enhancing effects of a mixture of Astragalus membranaceus (Fisch.) Bunge, Angelica gigas Nakai, and Trichosanthes Kirilowii (Maxim.) or its active constituent nodakenin. Journal of Ethnopharmacology, 285, 114893.

• Kim, M. C., Lee, G. H., Kim, S. J., Chung, W. S., Kim, S. S., Ko, S. G., & Um, J. Y. (2012). Immune-enhancing effect of Danggwibohyeoltang, an extract from Astragali Radix and Angelicae gigantis Radix, in vitro and in vivo. Immunopharmacology and Immunotoxicology, 34(1), 66-73.

• Park, Y. M., Lee, H. Y., Shin, D. Y., Kim, D. S., Yoo, J. J., Yang, H. J., ... & Bae, J. S. (2022). Immune-Enhancing Effects of Co-treatment With Kalopanax pictus Nakai Bark and Nelumbo nucifera Gaertner Leaf Extract in a Cyclophosphamide-Induced Immunosuppressed Rat Model. Frontiers in Nutrition, 9.

• Shin, H. J., Gwak, H. M., Lee, M. Y., Kyung, J. S., Jang, K. H., Han, C. K., ... & Kim, S. H. (2019). Enhancement of respiratory protective and therapeutic effect of Salvia plebeia R. Br. extracts in combination with Korean red ginseng. Korean Journal of Medicinal Crop Science, 27(3), 218-231.

Minor Concerns

1. Why was the extract diluted in 1% FBS and P/S and not in plain medium?

- The reaction of NO production after adding Griess reagent causes to form a pink-red azo dye, and the pink color is the result of nitic accumulation. It is not led by the color of the RPMI (normally red). Also, as you know, phenol red has own pink color. It can inhibit the identification of NO production in case phenol red is included. 

 - Therefore, the extract was diluted in RPMI-1640 medium without phenol red, 1% FBS, or P/S, which did not have color.

- Thanks. 

2. The authors need to state the concentration of the cells used for RNA extraction, Western blot as well as Macrophage phagocytosis assay.

- It was corrected.

- Thanks.

3. The authors did not state clearly how the mRNA expression analysis was done. They seem to use Actin as control but they did not say how many biological and technical replicates were done and how they analysed for mRNA expression.

- It was corrected.

- Thanks.

---

## [Decision Letter · Decision Letter 1]

8 Jan 2024

In vitro immune-enhancing effects of Platycodon grandiflorum combined with Salvia plebeian via MAPK and NF-κB signaling in RAW264.7 cells

PONE-D-23-29766R1

Dear Dr. Park,

We’re pleased to inform you that your manuscript has been judged scientifically suitable for publication and will be formally accepted for publication once it meets all outstanding technical requirements.

Kind regards,

Babatunde Olanrewaju Motayo, Ph.D.

Academic Editor

PLOS ONE

Additional Editor Comments (optional):

Reviewers' comments:

Reviewer's Responses to Questions

**Comments to the Author**

1. If the authors have adequately addressed your comments raised in a previous round of review and you feel that this manuscript is now acceptable for publication, you may indicate that here to bypass the “Comments to the Author” section, enter your conflict of interest statement in the “Confidential to Editor” section, and submit your "Accept" recommendation.

Reviewer #1: All comments have been addressed

Reviewer #2: All comments have been addressed

2. Is the manuscript technically sound, and do the data support the conclusions?

Reviewer #1: Yes

Reviewer #2: Yes

3. Has the statistical analysis been performed appropriately and rigorously? 

Reviewer #1: Yes

Reviewer #2: Yes

4. Have the authors made all data underlying the findings in their manuscript fully available?

Reviewer #1: Yes

Reviewer #2: Yes

5. Is the manuscript presented in an intelligible fashion and written in standard English?

Reviewer #1: Yes

Reviewer #2: Yes

6. Review Comments to the Author

Reviewer #1: Lines 106 – 109: Please state the exact volume of the supernatants and the WST solution that were added to each well

Lines 125 – 127: A brief general description of the ELISA procedure as requested in the first review was not effected?

Line 159: Please replace “amount” with ‘quantity’ or ‘volume’

Reviewer #2: None. The Authors have addressed the initial comments raised satisfactorily. They also ensured to reflect this in their manuscript

7. PLOS authors have the option to publish the peer review history of their article (what does this mean?). If published, this will include your full peer review and any attached files.

Reviewer #1: **Yes: **Dr. Kazeem S. Akinwande

Reviewer #2: No

---

## [Editor Report · Acceptance letter]

24 Jan 2024

PONE-D-23-29766R1 

PLOS ONE

Dear Dr. Park, 

I'm pleased to inform you that your manuscript has been deemed suitable for publication in PLOS ONE. Congratulations! Your manuscript is now being handed over to our production team.

Kind regards, 

on behalf of

Dr Babatunde Olanrewaju Motayo 

Academic Editor

PLOS ONE